# Multi-Megawatt Horizontal Axis Wind Turbine Blade Optimization Based on PSO Method

**Hamid R. Kaviani [1] and Mohammad Moshfeghi [2,\*]**

1   School of Mechanical Engineering, Malayer University, Malayer P.O. Box 14395-515, Iran
2   College of Engineering, Mathematics and Physical Sciences, University of Exeter, Exeter EX5 2FN, UK
\*   Correspondence: mmoshfeghi@sogang.ac.kr

**Abstract:** Blade optimization methods are crucial for wind turbine design. In this research, a new set of values for the parameters of the Particle Swarm Optimization (PSO) method is proposed, and its effects on the enhancement of the power generation of the NREL WP-Baseline 1.5 MW horizontal axis wind turbine are investigated. First, the PSO parameters are tuned, and the convergence speed and the optimal accuracy of the objective function are improved. Then, the Class/Shape Transformation (CST) method is employed, and an appropriate order of the shape function polynomial is selected. In the third step, the WP-Baseline 1.5 MW blade is optimized according to the tuned PSO parameters, and the airfoil is represented by CST algorithms. Later, a CFD model, including 37 million cells and an IDDES turbulence model, was validated and used for a comparison of the power generation of the original and optimized blades. The optimized blade produced more power for all wind speeds above 4.5 m/s, with a maximum of 13.8% at 10 m/s and +7.25% at the rated wind speed (11.5 m/s). It should be noted that since the algorithms, tunings, and techniques adopted in the present study were general, the presented method can be used as a systematic approach for the aerodynamics shape optimization of multi-megawatt HAWTs.

**Keywords:** aerodynamic optimization method; horizontal axis wind turbine; particle swarm optimization; class/shape transformation

## 1. Introduction

The aerodynamics of wind turbine blades can be improved using flow manipulation (active and passive flow control techniques [1,2]) or blade shape optimization [3,4]. The algorithms used in the blade optimization of the horizontal axis wind turbine (HAWT) blades are usually classified into two categories: Gradient-Based Algorithms (GBAs) and Metaheuristic Algorithms (MAs). Consequently, the optimal blade's shape depends on the algorithm implemented.

GBAs are mainly used because they are swift in achieving final geometry shapes. In addition, they can facilitate the implementation of a large number of optimization constraints, making this category appropriate for complex problems. However, they have robustness issues, especially when dealing with a large number of optimization variables [5]. In addition, they are sensitive to initial conditions [6] and may not converge to the global optimum [7].

MAs are often inspired by nature. One of the most popular MAs is the Genetic Algorithm (GA) [8]. The GA mimics Darwin's theory of survival and is usually reliable and robust, but it is time-consuming [5]. Introduced by Kennedy et al. [9], Particle Swarm Optimization (PSO) is another MA. A PSO algorithm simultaneously uses the social and individual intelligence of the population (e.g., birds, insects, etc.) to find the best value for the objective function (i.e., food). PSOs have shown more promising performances compared to some other MAs, such as the GA [10]. Moreover, Mirjalili et al. [11] showed the superiority of PSO for airfoils over the Non-Dominated Sorting Genetic Algorithm-II (NSGA-II) and Tabu Search, as two MA cases.

In addition, the geometric parameterization methods, such as Bezier [12], Non-Uniform Rational B-Splines (NURBS) [13], B-Spline [14], PARSEC [15], and Class/Shape Function Transformation (CST) [16], play an important role in the aerodynamic optimization of HAWTs.

According to the previous research, the CST is more efficient compared to several other techniques, such as Bezier curves and Hicks–Henne bump functions, and has an excellent exploratory characteristic [16]. Using five desirable characteristics, i.e., orthogonality, completeness, parsimony, intuitiveness, and flawlessness, Sripawadkul et al. [17] showed that the CST method is one of the best choices among some of the other geometry parameterization techniques (i.e., Ferguson's curves, Hicks–Henne bump functions, B-Splines, and PARSEC).

As discussed above, the blade optimization methods have always been important in wind turbine research. Following the previous research works, the purpose of the present research is to introduce a fast and optimal framework for the improvement of HAWT output power. To achieve this goal, parametric studies were performed for CST and PSO. In addition, the Improved Blade Element Momentum (IBEM) theory was used for the estimation of the objective functions (i.e., wind turbine power). Then, MATLAB was employed to integrate the IBEM theory, the geometric CST technique, and the PSO algorithm. This was followed with the use of the NREL 1.5 MW WP_Baseline for the optimization case. After optimization, transient IDDES simulations were performed, and the power performance of the new geometry was compared to the original one, proving the success of the proposed optimization method. A roadmap containing the details of these steps is demonstrated in Figure 1.

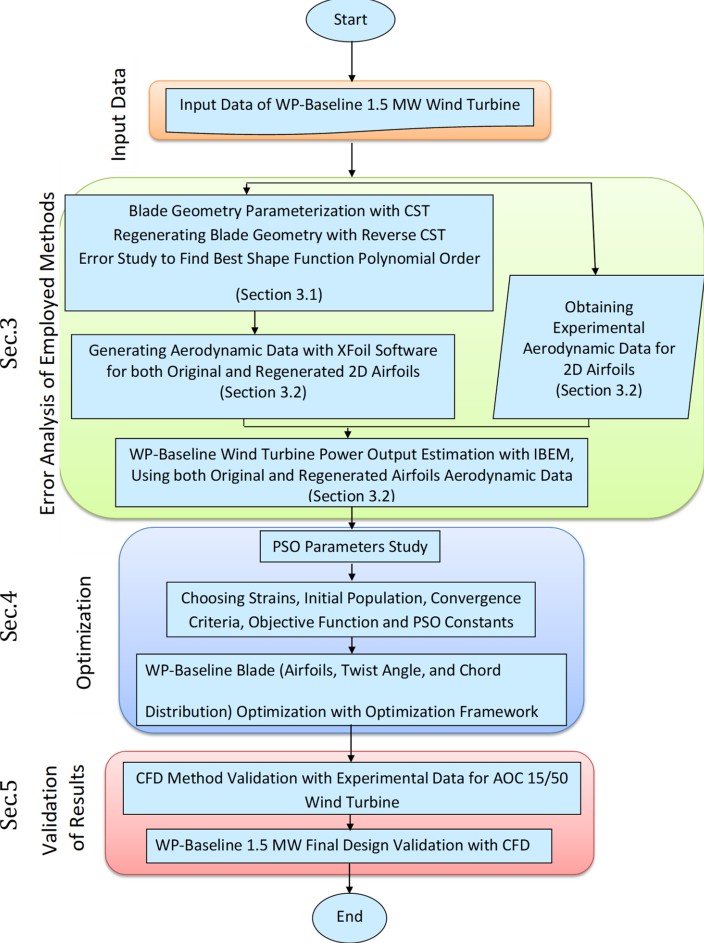

**Figure 1.** Flowchart of the optimization process.

## 2. Methodology

Three different tools are employed for the optimization phase:

- The CST technique for geometric parameterization (Section 2.1);
- The objective function evaluation tools (Section 2.2);
- The Single-Objective Particle Swarm Optimization (PSO) algorithm (Section 2.3).

### 2.1. The CST Technique

The geometric Class/Shape Function Transformation technique (CST) [18] is used in this study to parameterize the blade geometry. This method uses two polynomial curves for the top and bottom lines to represent an airfoil. As a result, it reduces the number of variables required to display the airfoil from at least 60 (number of points) to the number of polynomial coefficients (e.g., $n = 6$ or 7). Mathematically, the CST method is described as:

$$\xi = C(\psi).S(\psi) + \psi \Delta \xi \tag{1}$$

where $c$ is the length of the airfoil's chord, and $\psi$ and $\xi$ are non-dimensional coordinates in the x and y directions ($\psi = x/c$ and $\xi = y/c$). The functions $S(\psi)$ and $C(\psi)$ are the shape and class functions, respectively, and are described as:

$$S(\psi) = \sum_{r=0}^{n} A_i \frac{n!}{r! \, (n-r)!} (1 - \psi)^{n-r} \psi^r \tag{2}$$

$$C(\psi) = \sqrt{\psi}(1 - \psi) \tag{3}$$

where $n$ is the order of the shape function, and $A_i$ is the $i$th scaling coefficient.

### 2.2. The Objective Function Evaluation Tool

Different methods with varying fidelity and computational costs are usually implemented to evaluate the objective function during the aerodynamic optimization of HAWTs. The most implemented method is the Blade Element Momentum (BEM) theory, which is used for optimization purposes in particular and is reasonably accurate and very cost-effective [19–23].

Over time, some modifications have been applied to improve the precision of the results achieved by BEM. These include: (a) tip-loss correction and hub-loss correction; (b) Glauert correction; (c) the dynamic stall model; and (d) tower influence correction. In addition, the improved version of the BEM (also known as IBEM) requires the aerodynamic coefficients of the airfoils to estimate the power generation of an HAWT. It should be mentioned that the aerodynamic coefficients of the airfoils generated during the optimization process in the present research are calculated by XFoil 6.94 software [24].

### 2.3. Single-Objective Particle Swarm Optimization (PSO) Algorithm

In the present study, a modified version of the Single-Objective PSO algorithm is used for the aerodynamic optimization of the HAWT's blade. The method uses the following equations [25]:

$$v_{m,n}^{new} = v_{m,n}^{old} + C_1 \times r_1 \times \left( p_{m,n}^{local\ best} - p_{m,n}^{old} \right) + C_2 \times r_2 \times \left( p_{m,n}^{global\ best} - p_{m,n}^{old} \right) \tag{4}$$

where $v_{m,n}$ is the particle velocity in the $m$ and $n$ dimensions; $p_{m,n}$ is the particle position in the $m$ and $n$ dimensions; $p_{m,n}^{local\ best}$ is the best position achieved by a particle in the $m$ and $n$ dimensions; and $p_{m,n}^{global\ best}$ is the best position found by the entire swarm. Parameters $r_1$ and $r_2$ are random factors, and their values are between [0, 1]; $C_2$ is the social factor; and $C_1$ is the cognitive constant.

The new position of particles is calculated as:

$$p_{m,n}^{new} = p_{m,n}^{old} + w \times v_{m,n}^{new} \tag{5}$$

where $w$ is the inertia weight.

### 2.4. CFD Settings (for Validation Phase)

CFD simulations are used for the verification of the final optimized blade. To create a valid CFD model, first all the CFD settings are validated through the simulation of the AOC 15/50 HAWT using the IDDES turbulence model [26]. The validated CFD settings are then used for the verification of the optimized NREL WP-Baseline 1.5 MW blades. IDDES is an improved method based on the Delayed Detached Eddy Simulation (DDES). As a wall-modeled Large Eddy Simulation (LES), this model is able to accurately predict flow properties [27]. Moreover, as a DES-based model, for the near-wall calculations the model is more cost-efficient than LES. The CFD settings, investigated prior to the main simulations, included mesh sensitivity analysis, first grid spacing ($y+ < 1$), and timestep for this DES-based turbulence, according to the authors' previous experiences [28].

## 3. Validation of the Geometry Parametrization Technique and 2D Aerodynamic Data

### 3.1. Geometry Parameterization Using CST

In this section, the CST technique, as a geometric parameterization tool, is validated for the airfoils of the 1.5 MW NREL wind turbine (i.e., S818, S825, and S826).

During the CST process, the class function $C(x/c)$ is used for the class demonstration (e.g., airfoil), and the shape function $S(x/c)$ is used for the detailed representation of the airfoil (e.g., the airfoil leading edge radius, etc.). As the order of the polynomial in the shape function affects the accuracy of the CST, its influence on the airfoil representation must be studied. Considering that it is not easy to visually determine the best order of the shape functions, two criteria have been employed: 1—maximum absolute error and 2—mean error. The following relations have been employed to calculate those two criteria:

$$Maximum\ absolute\ error = Max.\left(\left|y_{i_{Ref}} - y_{i_{CST}}\right|_{i=1}^{n}\right) \tag{6}$$

$$Mean\ error = \frac{\sum_{i=1}^{n}\left|y_{i_{Ref}} - y_{i_{CST}}\right|}{n} \tag{7}$$

where $n$ is the total number of points used to represent the airfoil, and $y_i$ is the vertical coordinate (m) of point $i$. The subscript "$Ref$" indicates the coordinates obtained from the reference airfoil, and the subscript "$CST$" indicates the coordinates obtained from the $CST$ method. The study was performed on all three airfoils of the 1.5 MW turbine (S818, S825, and S826). The various orders of the shape functions from 3 to 9 were examined. The best orders for the shape functions and their corresponding errors are presented in Table 1.

**Table 1.** Error analysis of CST method.

| Airfoil | The Best Shape Function Order | Mean Error (%) | Max. Abs. Error (%) |
|---|---|---|---|
| S818 | 6 | 0.0881 | 0.37 |
| S825 | 7 | 0.0796 | 0.33 |
| S826 | 7 | 0.0795 | 0.22 |

As is shown in Figure 2, increasing the polynomials up to the order of 7 improved the accuracy of the regenerated airfoil. A further increase in the polynomial order caused bumpiness on the suction surface and reduced the accuracy of the airfoil representation.

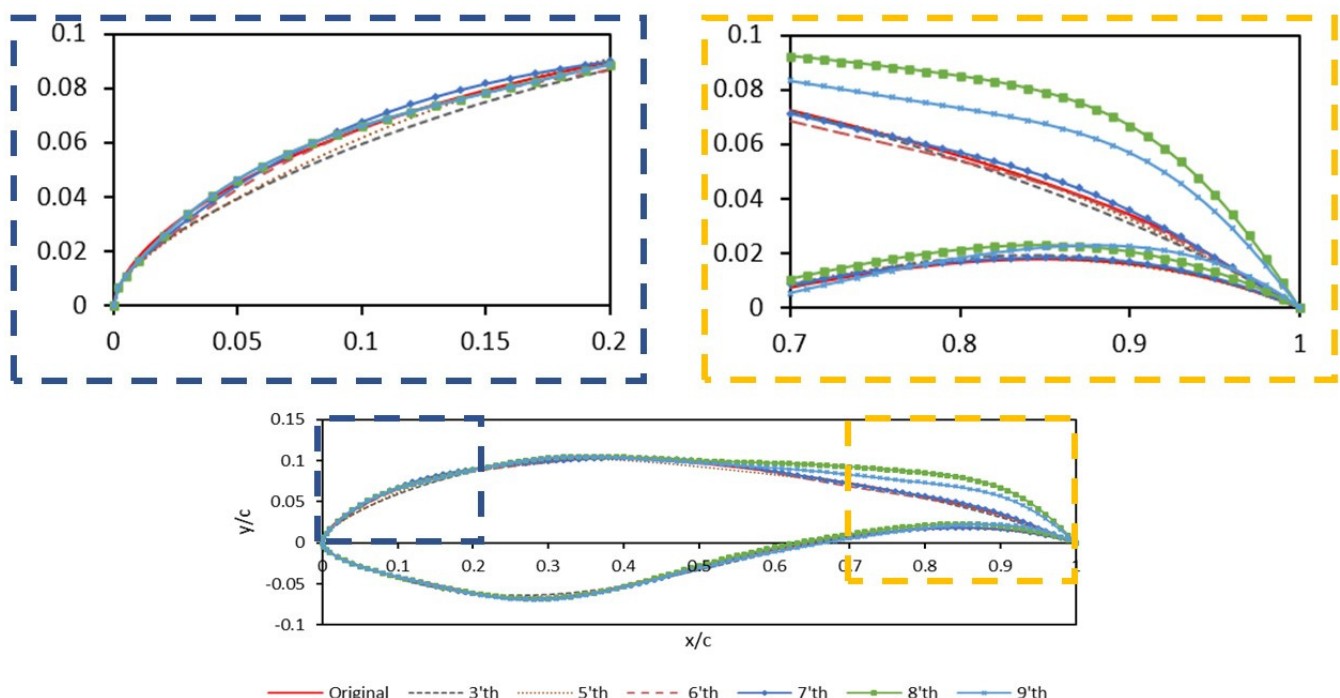

**Figure 2.** A comparison between the original S825 airfoil and those regenerated using CST function with different polynomial orders.

The results also showed that for the thicker airfoil, S818, a 6th-order polynomial was more accurate than the higher-order ones and avoided the bumpiness shown in Figure 2. The outcomes related to the S826 airfoil demonstrated that the best shape function is of the 7th order. Table 1 summarizes the errors of the aforementioned airfoils.

In addition to the shape of the airfoil, the blade chord length and the twist angle distributions were parameterized with a 6th-order Bézier curve; the 6th-order Bézier curve is equivalent to the CST method when the class function is set to unity, and the outcomes fitted well with the original curves. The maximum absolute errors for the chord and the twist representation were 0.1% and 0.22%, respectively.

*3.2. Airfoil Aerodynamic Data Validation*

The IBEM requires the airfoil's data at different radial sections in order to calculate the output power generated by a wind turbine blade (at each radial section). During the optimization process, new airfoils are generated, and the power prediction of a wind turbine needs their aerodynamics data. The XFoil package is used for this purpose.

Receiving the coordinates of the constituent points on an airfoil, the XFoil reproduces an airfoil's shape by line segments connecting the given points. Therefore, the more points on the airfoil, the more precisely the flow can be modeled. Usually, airfoils are represented by around 60 points on their upper and lower lines; however, to have a more accurate analysis, the present CST method used 200 points.

IBEM predicted the nominal power of the WP_Baseline as 1507.13 kW at a rotational speed of 20.5 rpm and a wind speed of 11.5 m/s [29]. Comparing this with the wind tunnel data, the regenerated airfoils by the CST method predicted the power values with an error equal to (0.5%), which is considerably smaller than that of the airfoil generated by XFoil (2.9%).

**4. Optimization of WP-Baseline Wind Turbine Blade**

The NREL's WindPACT_Baseline 1.5 MW HAWT was selected for the optimization in this study [29,30]. This turbine, hereinafter referred to as WP-Baseline, is a three-bladed HAWT with a variable rotational speed that gradually increases from 6 rpm (at a cut-in

speed of 3 m/s) to 20.5 rpm (at the rated speed of 11.5 m/s). Beyond the rated wind, the rotational speed is kept constant by changing the blades' pitch angle. The WP_Baseline turbine configuration and operational data are presented in Table 2.

**Table 2.** The WP_Baseline turbine configuration and operational data [29,30].

| | |
|---|---|
| Turbine version | 1.5A08C01V03cAdm |
| Rated power | 1500 kW |
| Rated wind speed | 11.5 m/s |
| Cut-in wind speed | 3 m/s |
| Cut-out wind speed | 25 m/s |
| Variable Speed | Cut-in to rated wind speed |
| Variable Pitch | Rated to cut-out wind speed |
| Rated rotational speed | 20.5 rpm |
| Coning angle | 0 degree |
| Tilt angle | 5 degrees |
| Hub height | 84.3 m |
| Rotor diameter | 70 m |
| Drivetrain efficiency | 95% |

In addition, the blade's aerodynamic specifications at different radial stations are presented in Table 3. During the optimization phase the maximum radius of the rotor and the operating condition of the turbine (rated wind and rotational speed) are set as the constraints of the optimization.

**Table 3.** The blade aerodynamic specifications at different radial sections.

| No | r (m) | Twist (deg) | Chord (m) | Airfoil |
|---|---|---|---|---|
| 1 | 5.07 | 11.1 | 2.27 | S818 |
| 2 | 7.29 | 11.1 | 2.59 | S818 |
| 3 | 9.50 | 10.41 | 2.74 | S818 |
| 4 | 11.72 | 8.38 | 2.58 | S818 |
| 5 | 13.94 | 6.35 | 2.41 | S825 |
| 6 | 16.15 | 4.33 | 2.24 | S825 |
| 7 | 18.37 | 2.85 | 2.08 | S825 |
| 8 | 20.59 | 2.22 | 1.91 | S825 |
| 9 | 22.80 | 1.58 | 1.75 | S825 |
| 10 | 25.02 | 0.95 | 1.58 | S825 |
| 11 | 27.24 | 0.53 | 1.42 | S825 |
| 12 | 29.45 | 0.38 | 1.27 | S825 |
| 13 | 31.67 | 0.23 | 1.129 | S826 |
| 14 | 33.89 | 0.08 | 0.98 | S826 |
| 15 | 35 | 0.0 | 0.90 | S826 |

Different objective functions have been employed for the aerodynamic optimization of the wind turbines, e.g., Annual Energy Production (AEP) [31], Cost of Energy (COE) [21,22], and power output [20]. As both the AEP and the COE estimations require site wind speed distribution data, those objective functions are used for site-specific optimization. However, power output estimation only needs the HAWT rated speed, and it was employed for the verification of the optimization methodology in this study.

### 4.1. PSO Parametric Study for Blade Optimization

The PSO algorithm is inspired by the behavior of swarms of flying birds. Each bird corrects its path and speed according to its own previous information and the other birds' information in order to find more food. This correction is conducted so that all the birds eventually gather at the location where the most food exists (convergence at the optimal point). The PSO is a population-based method; hence, the key parameters of the PSO algorithm are:

- Choosing an efficient and cost-effective number of birds to cooperate in the food search;
- Selecting the search space boundaries (for food);
- Appropriate speed of birds for food search;
- Convergence criteria for final location of birds (location of maximum food);
- Effects of individual knowledge (influenced by the cognitive constant, $C_1$);
- Experience gained from the community (influenced by the social factor, $C_2$).

In this study, the objective function of optimization is the power of the HAWT (equivalent to the food in a bird problem). The birds are equivalent to the CST polynomials, and the location of the maximum food is the optimum shape of the blades (airfoil, chord, and twist angle).

One weakness of the population-based algorithms is the convergence of particles (birds) at local optimal points, while a better convergence means that birds should gather at a global optimal point (the highest value of the objective function in the whole search interval). This is the reason that a factor called inertial weight (i.e., "$w$" in Equation (5)) is employed. The weight of inertia at the beginning of the optimization is set large enough for the population particles (birds) to be well distributed over the search space. Then, as the optimal global range is found, the weight of inertia is reduced to lessen the search speed and increase the search accuracy. This eventually results in a precise location of the maximum food (equivalent to finding the best blade shape for the maximum power in an HAWT).

The present research contained a parametric study of the PSO on the number of birds (different variations of the blade geometry), the size of the search space (deviation of CST parameters from their initial values), the convergence criterion (accuracy of convergence), the inertia weight, the cognitive constant, the social factor, and the velocity clamping factor. To start with, first an optimization was performed on the twist angle of the 1.5 MW HAWT blades, and then, the tuned parameters were used for the optimization of the chord length and airfoil shape. The optimization constraints were defined as:

- $\frac{\partial\ Twist\ angle}{\partial\ Radius} \geq 0$
- *Twist angle* $\geq -2$ *degrees*

The first study was performed to investigate how the number of birds (different variations of the blade geometry) affects wind turbine power improvement. Each bird (variation of the blade geometry) contained seven constants needed for the twist angle representation with Bézier polynomials (Figure 3).

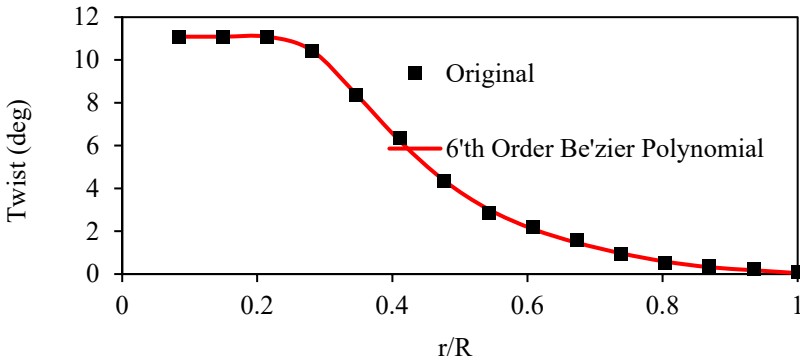

**Figure 3.** Representation of the blade twist using the 6th-order Bézier curve.

The convergence criterion for the termination of the optimization procedure is $10^{-3}$. The inertia weight decreased from $w = 0.9$ at the beginning to $w = 0.4$ at the end, and the cognitive constant was set to $C_1 = 2$ [32]. In this study, the HAWT power output is the objective function (equal to food availability), and the variation of the blade geometry (number of birds) affects the power enhancement. It is important to note that in order to eliminate the influence of random function on the optimization results, the optimizations are repeated three times for bird numbers from one to ten.

As shown in Figure 4, at least two birds (variations of the blade geometry) were needed for the optimization to reach the 1% power increment. The results also showed that there is no reason to exceed the number of birds beyond six.

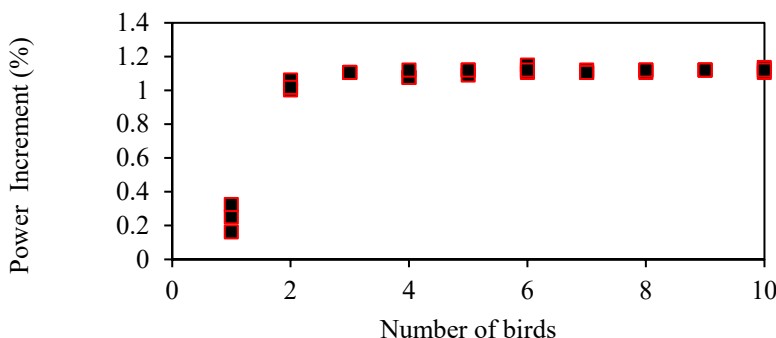

**Figure 4.** Effects of the number of birds on output power.

Moreover, the effect of the number of birds on the convergence time has been studied. As shown in Figure 5, if the number of birds exceeds seven, the convergence time rises exponentially.

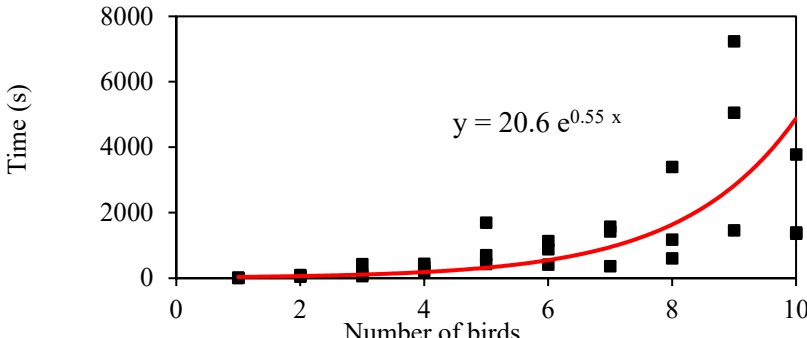

**Figure 5.** The effect of the number of birds on convergence time (e = 2.71828 is Euler's number).

In addition, the effect of the velocity clamping factor (VCF) on convergence time is presented in Figure 6. Increasing the velocity clamping factor does not seem to be useful with regard to convergence time.

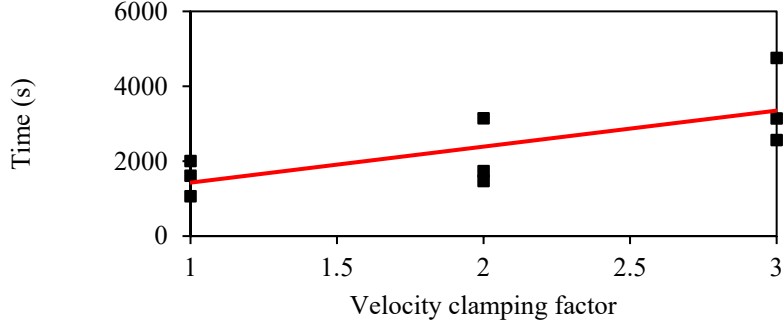

**Figure 6.** The effect of velocity clamping factor on convergence time.

Furthermore, a study on the effects of the cognitive constant and the minimum inertia weight on the convergence time revealed that the minimum inertia weight of $w = 0.7$ accompanied by a cognitive constant of $C_1 = 2$ gives the best result (29 kW power enhancement; see Figure 7). This study also showed that the minimum inertia weight above

$w = 0.7$ resulted in convergence of the local optimums (blue points in Figure 7), which were not desirable situations.

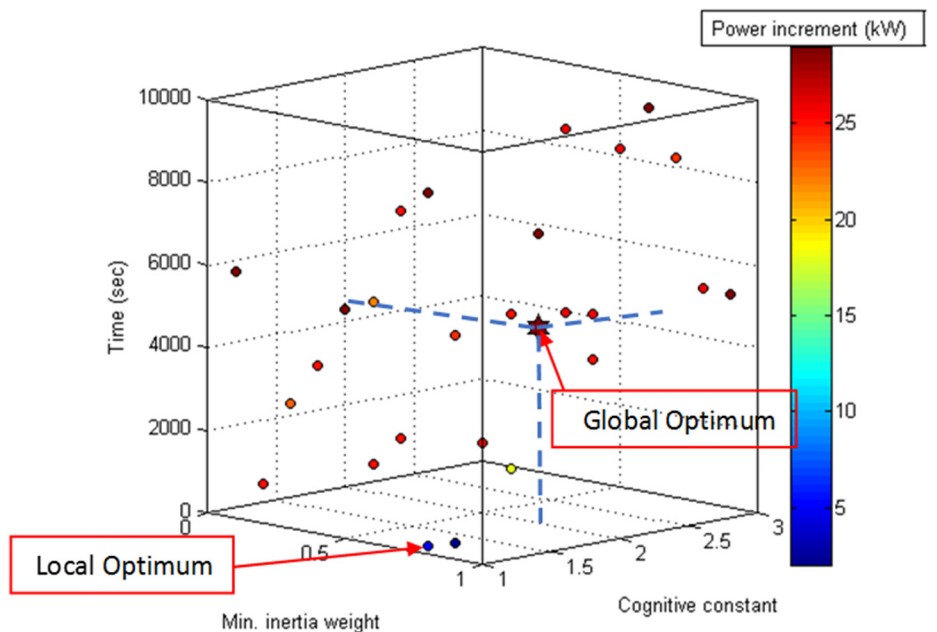

**Figure 7.** Cognitive constant and min. inertia weight study.

Furthermore, it was observed that by decreasing the convergence criterion from $10^{-3}$ to $10^{-4}$ the computation time was tripled while the power increased by only 0.05%. Hence, the value of $10^{-3}$ was decided on as a computationally efficient choice.

Moreover, for investigating the effect of the size of the search space (SSS) on the power optimization, the variation of the initial values of SSS from $\pm10\%$ to $\pm100\%$ (for the 6th-order Bézier curve coefficients) was examined (Figure 8). For the absolute variation of 90% and above, the twist angle increment near the hub led to a 3.8% power enhancement. This was because the increase in the twist angle was directly transferred to the local angle of attack.

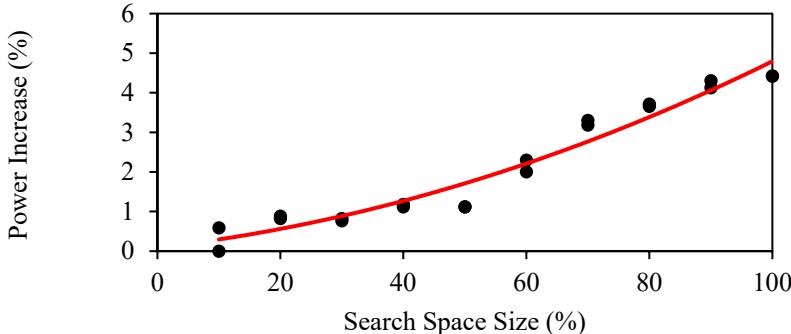

**Figure 8.** The effect of search space size on power increment.

Furthermore, the IBEM revealed that the twist optimization resulted in a maximum power enhancement of about 57 kW (3.8%), as compared to the baseline wind turbine (Figure 9).

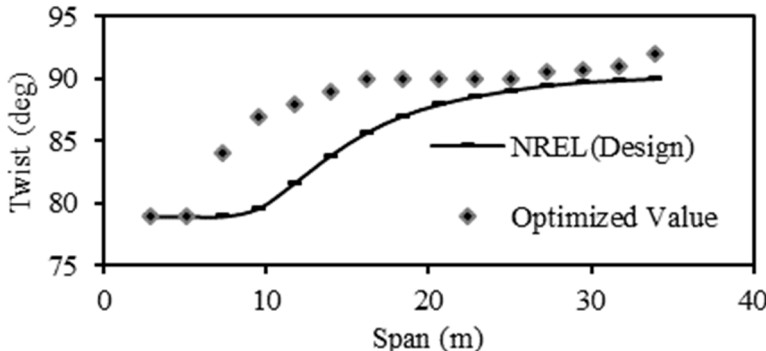

**Figure 9.** The baseline and the optimized twists.

To wrap up this part, a summary of the proposed parameters for the PSO algorithm is presented in Table 4. The social factor, $C_2$, is assumed to be equal to $(4 - C_1)$, as proposed by Clerc and Kennedy [33]. The HAWT power increment achieved by optimizing each airfoil is presented in Table 5.

**Table 4.** The proposed values for PSO parameters.

| Number of Birds | Search Space Size % | $w_{min}$ | $C_1$ | $C_2$ | $v_{max}$ | Convergence Criterion |
|---|---|---|---|---|---|---|
| 6 | 90 | 0.7 | 2 | 2 | 2 | $10^{-3}$ |

**Table 5.** The HAWT power increment by optimizing airfoils.

| Airfoil Used for the Blade | Power Enhancement by Each Section with the Optimized Airfoil (kW) |
|---|---|
| Optimized S818 | 3.9 kW |
| Optimized S825 | 16.7 kW |
| Optimized S826 | 19.6 kW |
| Total | 40.2 kW |

*4.2. Airfoil Optimization*

The NERL 1.5 MW wind turbine blade uses the S818, S825, and S826 airfoils. In the present research, the maximum thickness-to-chord ratio was used as a constraint for the optimization of the airfoils, such as:

Thickness-to-chord ratio of an optimized airfoil $\leq$ the original airfoil's thickness-to-chord ratio.

The proposed values for the PSO parameters (Table 4) were used for the airfoils' optimization. Based on the IBEM results, the airfoils' optimization enhanced each airfoil's behavior as well as the wind turbine's power generation by 2.68% (see Table 5). The original and the optimized geometry of the airfoils are represented in Figures 10–12.

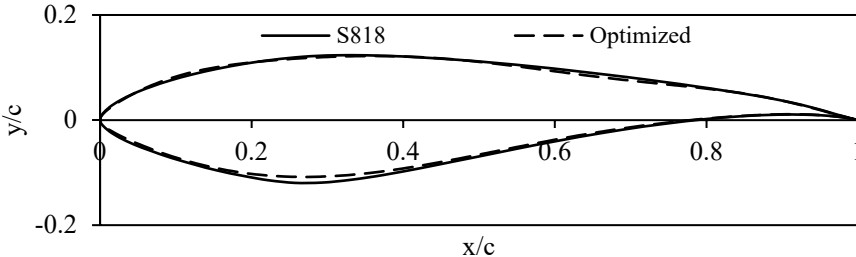

**Figure 10.** The original and the optimized S818 airfoil shape.

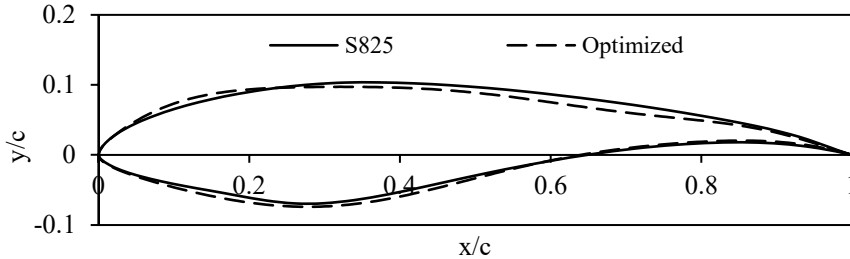

**Figure 11.** The original and the optimized S825 airfoil shape.

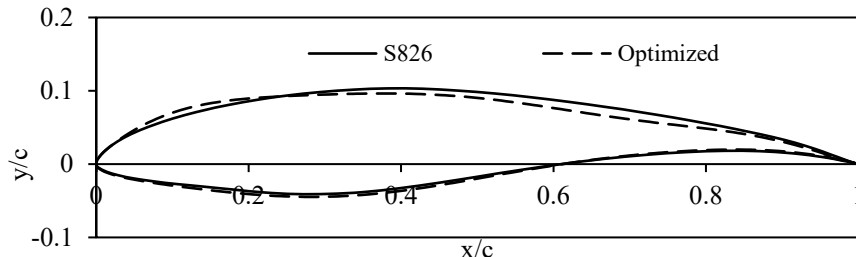

**Figure 12.** The original and the optimized S826 airfoil shape.

*4.3. Chord Optimization*

Furthermore, chord optimization was performed. Due to the structural considerations of a blade, two constraints were applied to the chord optimization process:

- Maximum chord length of the optimized blade $\leq$ original blade's maximum chord;
- $\frac{\partial\ Chord}{\partial\ Radius} \leq 0$

Figure 13 represents the original and optimized chord length values at different radial positions along the blade span. The optimized chord increased the power output by 29.9 kW (2%).

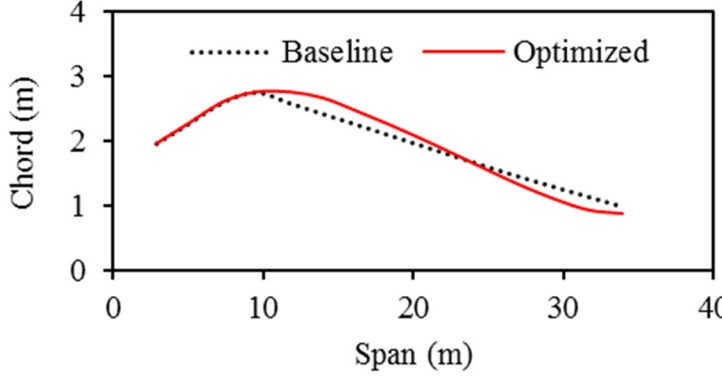

**Figure 13.** The baseline and the optimized chord distributions along the blade span.

## 5. Validation of the Optimized Blade

To evaluate the optimizations proposed, verification of the results was conducted using a two-stage process. In the first stage, the CFD settings (including mesh resolution, timestep, and y+) were investigated so that the results of the IDDES simulations with those parameters could be validated by the experimental results of the AOC 15/50 (50-kW) HAWT. Then, the same CFD settings were used for the CFD simulation of the optimized NREL WP_Baseline. As will be shown later in this section, the optimized geometry enhanced the power generated by the turbine by about 7%.

### 5.1. Validation of CFD Settings

The AOC 15/50 turbine was used for the validation of the used CFD setting. The information about the AOC 15/50 turbine is presented in Table 6. The ANSYS-FLUENT commercial package was employed for the CFD calculation. The time step for the IDDES simulation was chosen and was equal to $\Delta t = 10^{-4}$ [s]. The output power was calculated by averaging the values of about one complete rotation (i.e., 3 s).

**Table 6.** Information on the operation and geometry of the AOC 15/50 turbine [34].

| | |
|---|---|
| Turbine version | 60 Hz |
| Rated wind speed | 12 m/s |
| Number of blades | 3 |
| Rotor diameter | 15 m |
| Operational wind speed | 4.9–22.3 m/s |
| Rotational speed (constant) | 65 rpm |
| Rotor position | Downwind |
| Coning angle | 6 degrees |
| Pitch angle | 1.54 degrees (toward feather) |
| Hub height | 25 m |

In Figure 14, the CFD domain's shape and size are depicted. The mapping technique, which improves the final mesh quality, was implemented for the meshing of the blade surfaces. For further increasing the accuracy of the results near the leading and trailing edges, a mesh refinement was applied for this area (Figure 15). Furthermore, by selecting the height of the first layer to be equal to $5 \times 10^{-6}$, the condition of y+ < 1 was achieved (Figure 16). Finally, a mesh independence study (Figure 17) showed that there was no need to increase the mesh elements number beyond about 16 million (for the AOC 15/50 turbine).

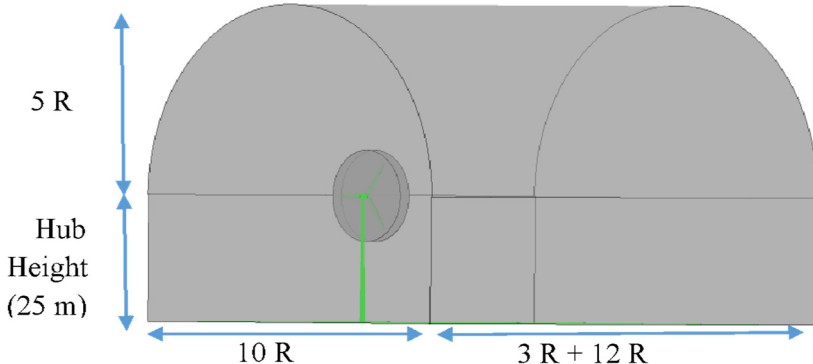

**Figure 14.** Domain topology and size.

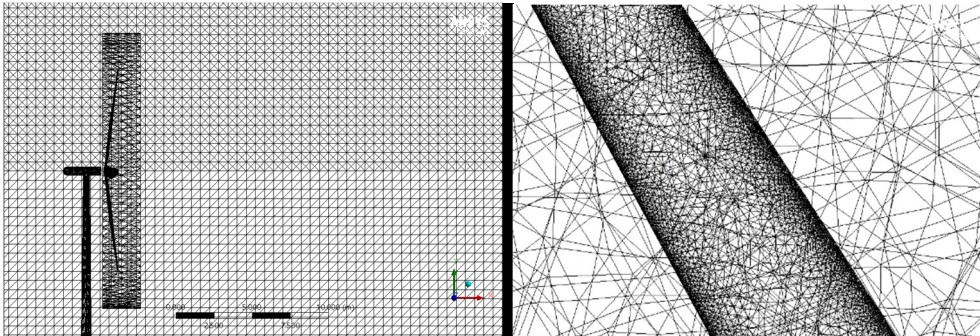

**Figure 15.** Fluid field mesh (**left**) and implementation of surface mapping technique on the blade surface (**right**).

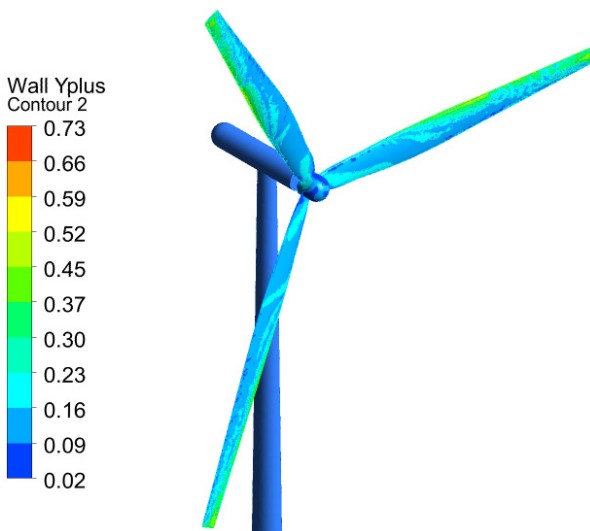

**Figure 16.** The distribution of y + 1 on the suction side (**left**) and the pressure side (**right**) of the AOC 15/50 blade.

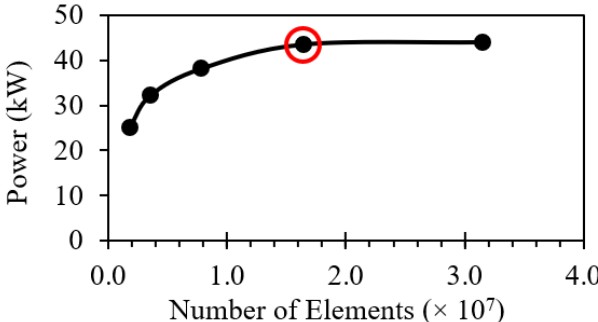

**Figure 17.** Mesh independence study for the AOC 15/50 blade rated power calculation (the red circle shows the chosen number).

The inlet and outlet of the domain were defined as the constant velocity and pressure, respectively. The power law was implemented for the wind profile modeling in the earth boundary layer and defined as [35]:

$$V(z) = V_{hub} \left( \frac{z}{z_{hub}} \right)^{\propto}, \quad \alpha = 0.2 \tag{8}$$

where $z$ is the height above ground level. The coupled algorithm was selected to couple the velocity and the pressure equations. The cell-based least squares method was used to spatially discretize the gradients. For the spatial discretization of the specific dissipation rate, pressure, and turbulent kinetic energy, the third-order scheme was employed. The air density was selected to be equal to the average air density of the site ($\rho = 1.012$ kg/m$^3$) [34]. To satisfy the Courant–Friedrichs–Lewy condition (CFL < 1.0) for the CFD simulation (Figure 18), the timestep was chosen as $\Delta t = 10^{-4}$ [s]. After the simulations were performed, the iso-surface plot of the Q-criterion (Figure 19) was checked to investigate the fineness of the volume cells near the blade and the appropriateness of the timestep and convergence criterion. The presence of small vortices in Figure 19 proves the appropriateness of the aforementioned parameters.

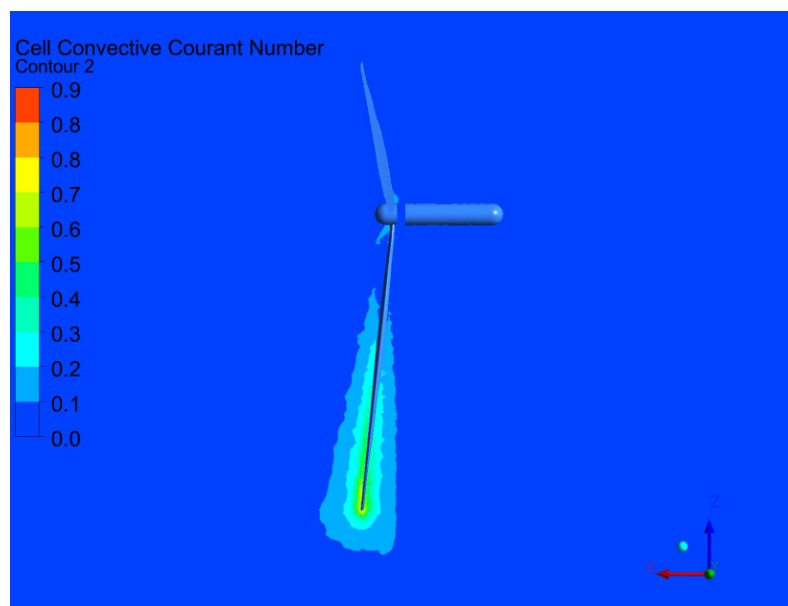

**Figure 18.** The Courant–Friedrichs–Lewy (CFL < 1) condition satisfaction on the flow field.

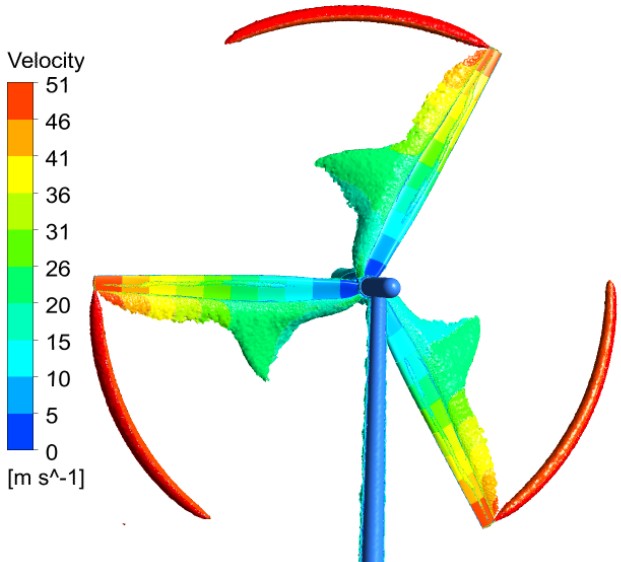

**Figure 19.** The iso-surface plot of Q-criterion.

In order to investigate the accuracy of the settings used in the previous CFD model, the power values calculated by the CFD method were compared with the experimental data of the AOC 15/50 turbine output power [34]. In Figure 20, the CFD values were modified according to the drive train efficiency of a typical fixed-pitch wind turbine according to [36]. As can be seen in Figure 20, the trend and point values of the computed values and the experimental data are in very good agreement with each other. These prove that the employed CFD settings were appropriately set.

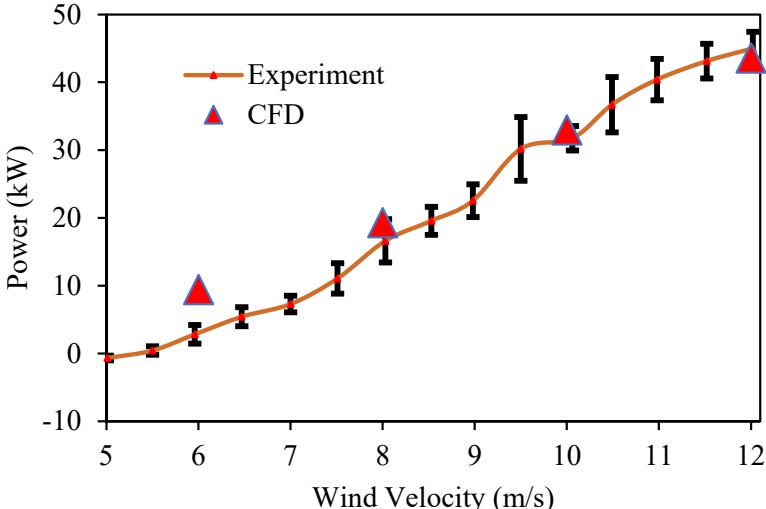

**Figure 20.** The accuracy of CFD results against the experimental values of AOC 15/50 wind turbine power output.

### 5.2. Verification of the Present Optimization Method

In this part, all the CFD settings (i.e., domain shape, meshing strategy, mesh resolution, y+, turbulence model, and timestep) that were validated in the previous section are adjusted to the NREL 1.5 MW and then applied to its CFD simulations with the optimized blade. It should be noted that the domain sizes were scaled with a factor of ($35 \div 7.5$), which is equal to the ratio of the radius of two blades.

Moreover, to have an independent mesh, a new mesh independence study was performed showing that 31 million cells were required for a valid CFD mesh of NREL 1.5 MW (Figure 21). In addition, the y+ < 1 condition, was re-checked over the blades' suction and pressure surfaces (as shown in Figure 22).

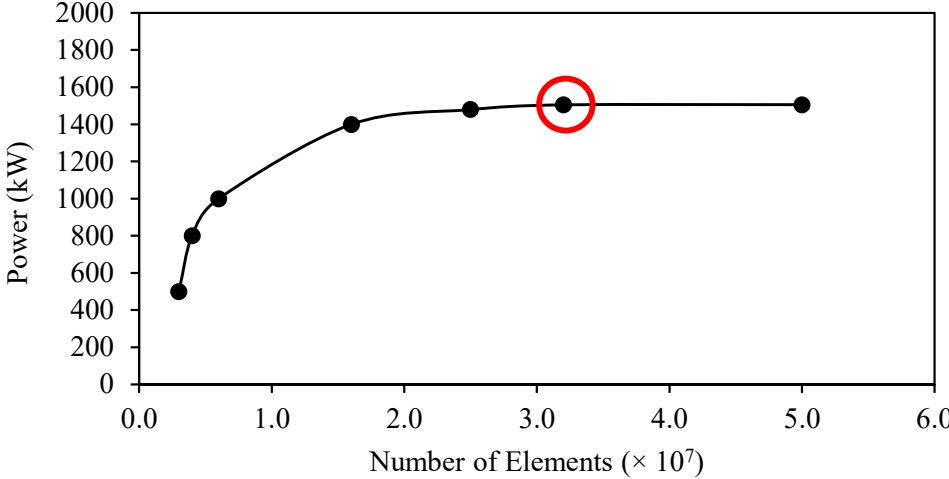

**Figure 21.** Mesh independence study for the WP_Baseline simulation (the red circle shows the chosen number).

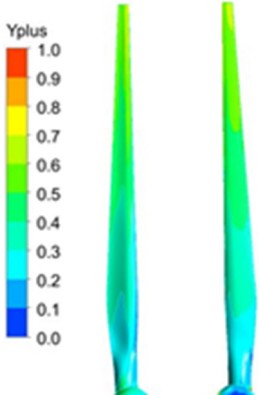

**Figure 22.** The satisfaction of y+ < 1 condition on the suction side (**left**) and pressure side (**right**) of the WP-Baseline 1.5 MW blade.

Figure 23 shows the grid generated for the CFD calculations around the 1.5 MW turbine. To improve the mesh, the mapping technique with a bias factor of 11 was applied on the blade surface (Figure 24).

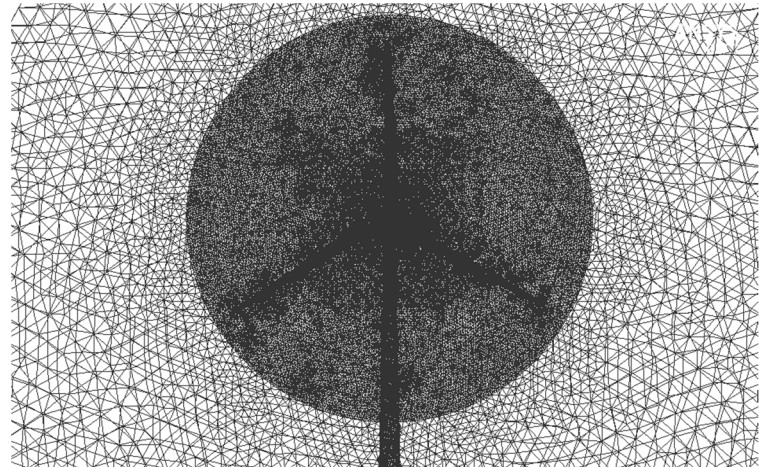

**Figure 23.** The grid generated for the CFD calculations around the 1.5 MW turbine.

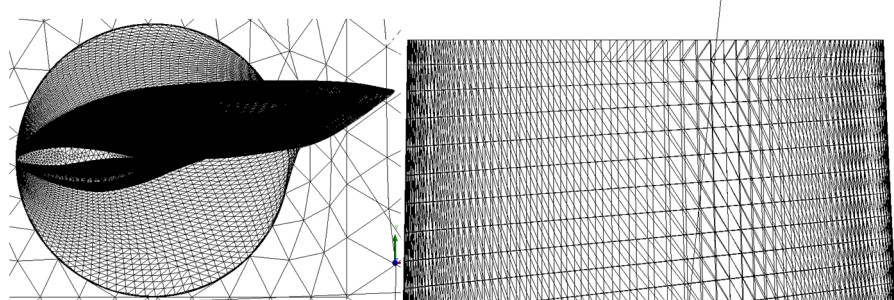

**Figure 24.** Execution of mapping technique with a bias factor of 11 on the blade surface.

The turbulence length scale was calculated according to the IEC 61400-11 standard [35], and the turbulence intensity was set to 5%. As in Section 5.1, the standard values of air density and viscosity at sea level were selected for the CFD simulations. The 1.5 MW turbine has a control system that serves to keep the rotor operating at the specified Tip

Speed Ratio (*TSR*) [30]. The *TSR* is defined as the linear speed of the blade's tip, normalized by the incoming wind speed:

$$TSR = \frac{R\Omega}{V_0} \tag{9}$$

where $V$ is the wind speed (m/s), $R$ is the blade span (m), and $\Omega$ is the rotational speed (rad/s). For the rated wind speed of 11.5 m/s, the rated rotational speed of 20.5 rpm, and the blade span of 35 m, the *TSR* is equal to 6.53. Hence, before the rated speed, the rotational speeds are calculated according to the relation $TSR = \frac{R\Omega}{V_0} = 6.53$. The flow vortices are shown in Figure 25 using the iso-surfaces of the Q-criterion (equal to 22 s$^{-2}$).

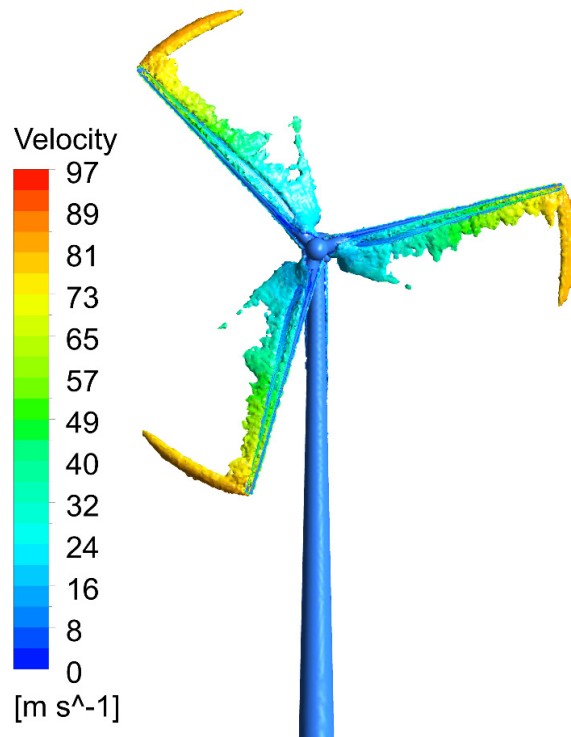

**Figure 25.** Iso-surfaces of the Q-criterion (22 s$^{-2}$).

In geometric–aerodynamic optimization, the objective function is assigned according to the performance expected from the machine. For example, the objective function for the optimization of the airplane wing is to increase the lift coefficient and decrease the drag coefficient [37]. The expectation from a wind turbine is to harvest wind energy. For this reason, in this research, increasing the output power is chosen as the objective function. To increase the output power, it is necessary to increase the torque of the blade (at the same wind speed). Torque is equal to the force multiplied by the lever arm, and the force is equal to the pressure difference multiplied by the blade area. Therefore, in Figures 26 and 27 the increase in pressure difference between the two sides of the blade is examined.

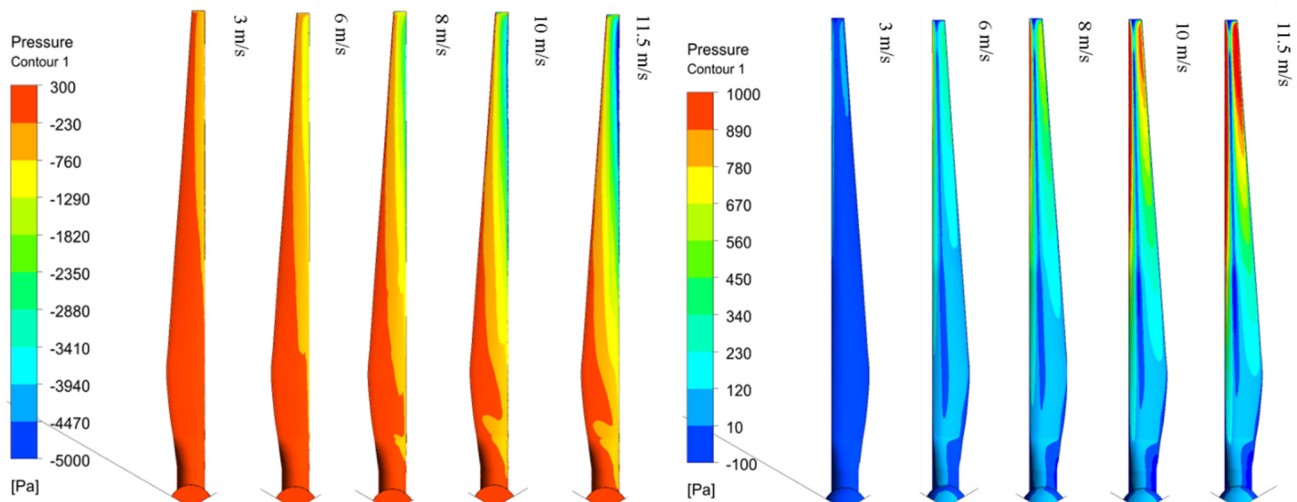

**Figure 26.** The 1.5 MW blade pressure contours on the suction surface (**left**) and pressure surface (**right**).

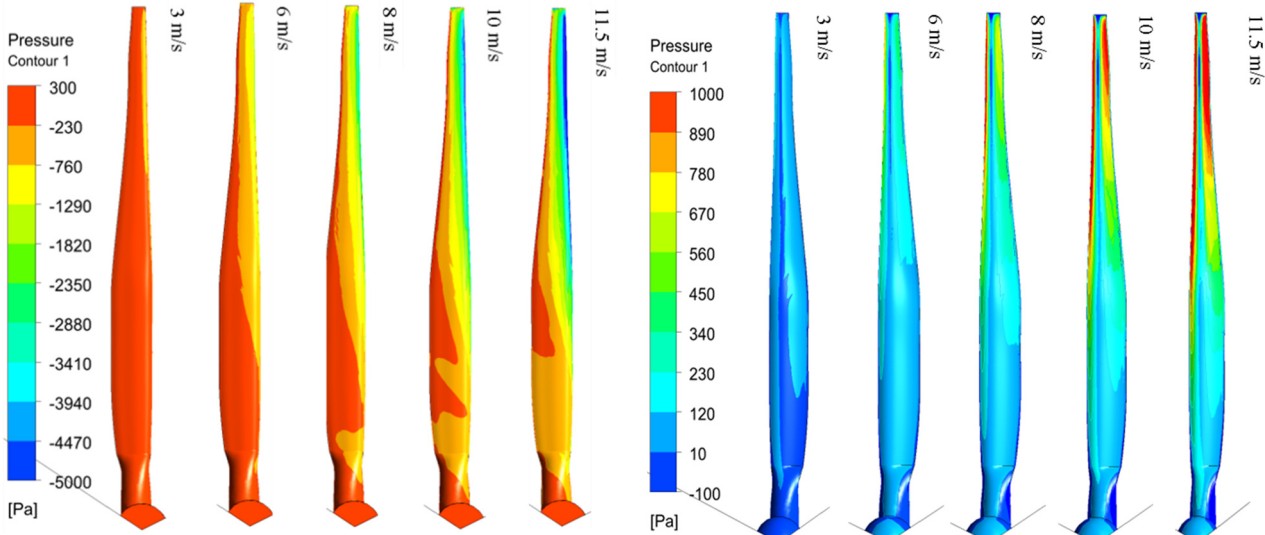

**Figure 27.** The optimized blade pressure contours on the suction surface (**left**) and pressure surface (**right**).

Moreover, an increase in the surface of the blade can increase the torque. Due to the intelligence of the PSO algorithm, this algorithm tends to increase the chord and the radius of the blade. Due to structural and cost concerns, the maximum chord length and rotor radius were chosen as constraints. However, in Figures 26 and 27 a slight increase in the chord of the blade (especially in its middle part) can be seen.

In Figure 26, the pressure contours for the baseline blade are depicted. With increases in local wind speed, the pressure at the lower surface of the blade increases, and at the upper surface, it decreases. As a result of increasing the pressure difference between the two sides of the blade, more rotational torque was produced, which led to an increase in the output power. Moreover, Figure 27 shows the pressure contour on the optimized blade surfaces. As shown, the pressure side of the optimized blade experiences higher pressure values than the original blade. On the other hand, the pressure values on the suction surface of the optimized blade are reduced. Hence, the optimized blade produced more rotational torque and power (except for 3 m/s velocities).

Figure 28 shows the pressure distribution at different radial sections. As can be seen, the pressure surface of the optimized turbine faces a trivial but almost uniform pressure increase. On the suction surface, however, the pressure fluctuates irregularly, and the effects of the optimization can be seen in the first 25% and the last 20% of the chord length from the LE. As a result, the enclosed area inside the pressure coefficient diagram of the optimized blade is larger than the original one.

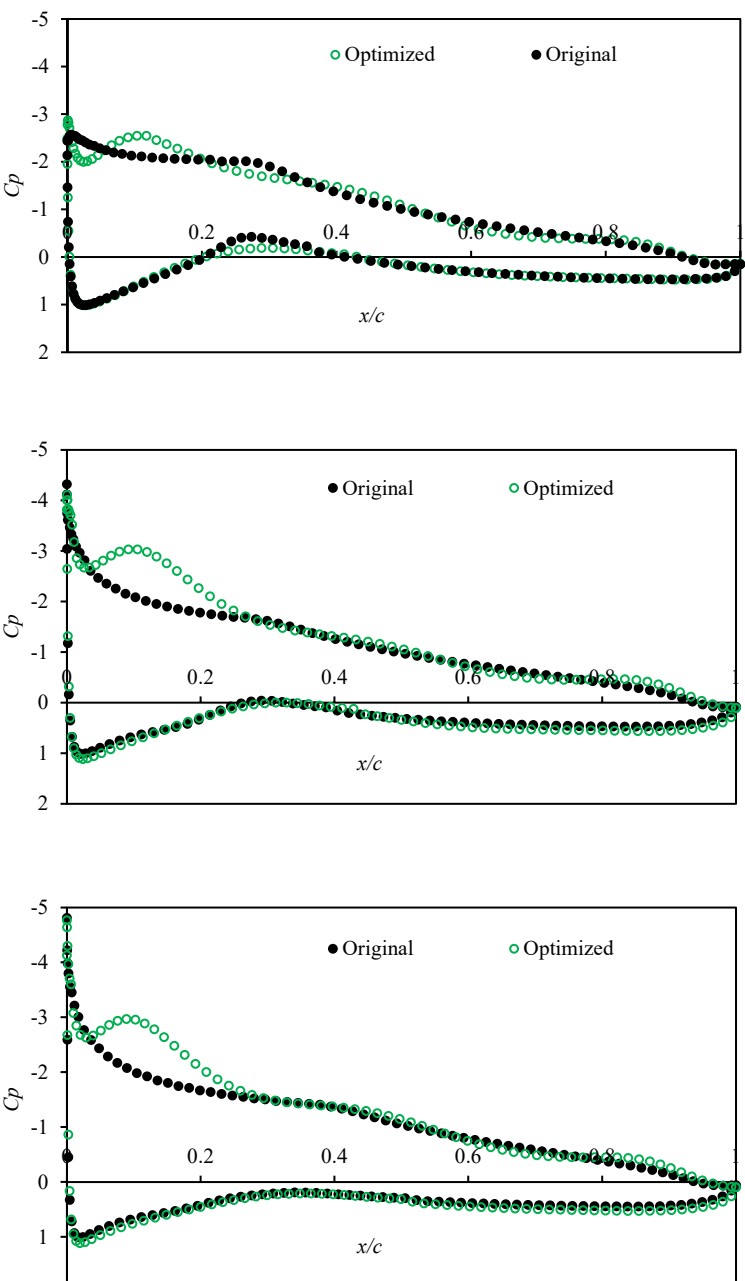

**Figure 28.** Pressure distribution at 35% (up), 65% (middle), and 95% of the blade span.

Although the optimization was performed for the rated speed, the study was further expanded to other wind speeds. Considering the efficiency of the drivetrain (=95%), the average power values of the turbine are presented in Table 7.

**Table 7.** The original and optimized WP_Baseline HAWT power for different air speeds.

| Wind Speed, [m/s] | Baseline Rotor Torque [kN.m] | Baseline Power [kW] | Optimized Rotor Torque (CFD) [kN.m] | Optimized Power (CFD) [kW] |
|---|---|---|---|---|
| 3 | 12.3 | 26.4 | 11.1 | 23.8 |
| 6 | 102.9 | 220.8 | 113.1 | 242.7 |
| 8 | 254.4 | 545.8 | 284.5 | 610.4 |
| 10 | 487.2 | 1045.4 | 554.7 | 1190.3 |
| 11.5 * | 725.7 | 1557.1 | 778.3 | 1670.0 |

*: rated speed.

Table 7 reveals that the optimized turbine considerably enhances the power production of the turbine over a wide range of wind speeds, and power increases are observed at different wind speeds. In addition, the changes in the generated power against the wind speed are plotted in Figure 29. As can be seen, except for wind speeds lower than 4.5 m/s, the optimized blade generates more power than the original blade. It should be mentioned that since this is a pitch-controlled wind turbine, for wind speeds above 11.5 m/s the control system changes the blade's pitch angle in such a way that the generated power remains constant.

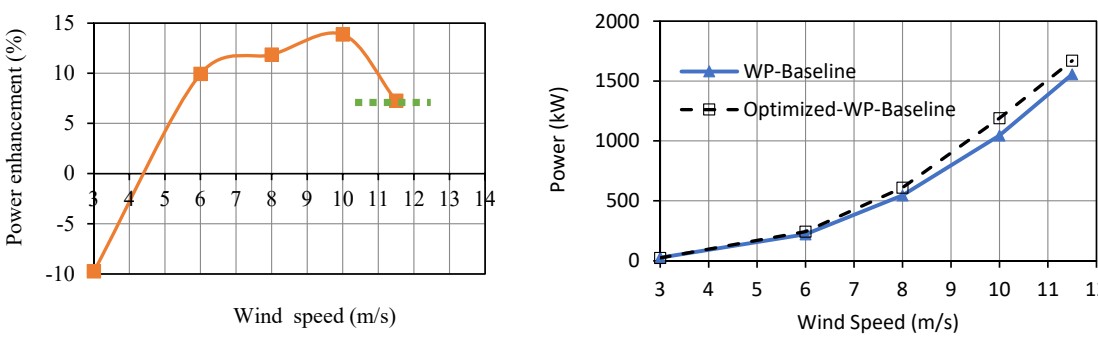

**Figure 29.** Power enhancement at different wind speeds.

## 6. Conclusions

In the present research, a PSO-based method is proposed and its effects on the power generation of the blades of the NREL WP-Baseline 1.5 MW horizontal axis wind turbine are verified. During the research, first the parameters of the PSO algorithm were tuned so that the convergence speed and the optimal accuracy of the objective function were improved. The resulting tuned parameters included the number of birds ($N$ = 6), the search space size (90% of original volumes), the minimum inertia weight ($w_{min}$ = 0.7), the cognitive constant ($C_1$ = 2), the velocity clamping factor ($v_{max}$ = 2), and the convergence criterion ($10^{-3}$). Then, the *CST* method, as a geometry parameterization technique, was employed for the optimization of the turbine's airfoils (S818, S825 and S826). The investigation of the shape function revealed that the polynomials of the order of $n$ = 6 for S818 and $n$ = 7 for S825 and S826 had minimal deviation from the original airfoils. In the third step, the geometry of the WP-Baseline 1.5 MW blade was optimized according to the tuned PSO parameters and the airfoil shapes regenerated by the CST algorithms.

Later, a CFD domain's shape, size, and settings (including mesh independency and the height of the first grid layer, y+ < 1) were investigated using the IDDES turbulence model in order to reach an accurate framework for the present CFD simulations by using the experimental results of the AOC 50/15 HAWT. In the final step, the validated framework and CFD settings were applied to a new set of CFD simulations for the original and optimized geometry of the NREL WP_Baseline. Using the CFD results of the NREL WP_Baseline, the final results confirmed that the optimized geometry of the turbine produced more power for all wind speeds greater than 4.5 m/s. More specifically, the optimized blade generated 7.25% more extra power than the original one at the rated wind speed (11.5 m/s).

　　　　Based on the promising results of the present optimization method, and since the PSO algorithms, their tunings, and the CST techniques used in the present study are general, one can conclude that this optimization method can be employed as a systematic approach for the aerodynamics shape optimization of multi-megawatt HAWTs.

**Author Contributions:** H.R.K., M.M. have equally contributed to the preparation of this article including "Conceptualization, methodology, software, validation, formal analysis, visualization and writing of the original and the revised paper". All authors have read and agreed to the published version of the manuscript.

**Funding:** This research received no external funding.

**Data Availability Statement:** Not applicable.

**Conflicts of Interest:** The authors declare no conflict of interest.

## Abbreviations

**Nomenclature**

| | |
|---|---|
| $A_i$ | the $i$th scaling coefficient |
| $C_1$ | cognitive constant [-] |
| $C_2$ | social factor [-] |
| C | chord length [m] |
| $C_D$ | drag coefficient [-] |
| $C_L$ | lift coefficient [-] |
| $C_p$ | pressure coefficient [-] |
| $C(\psi)$ | class function value at $\psi$[-] |
| $p_{m,n}$ | particle position in m and n dimensions [-] |
| $r_1, r_2$ | random factors [-] |
| $S(\psi)$ | shape function value at $\psi$[-] |
| $v_{m,n}$ | particle velocity in m and n dimensions [m/s] |
| W | inertia weight [-] |
| X | chord-wise coordinate [-] |
| $y^+$ | dimensionless wall distance [-] |
| Y | vertical coordinate [m] |

**Greek symbols**

| | |
|---|---|
| M | viscosity [kg/m.s] |
| P | density [kg/m$^3$] |
| $\psi$ | non-dimensional chordwise coordinate (=x/c)[-] |
| $\xi$ | non-dimensional vertical coordinate (=z/c)[-] |
| $\Delta Z$ | closure thickness [m] |

**Superscripts**

| | |
|---|---|
| local_best | the best quantity achieved by a particle in PSO algorithm |
| global_best | the best quantity achieved by the entire swarm in PSO algorithm |

**Subscripts**

| | |
|---|---|
| LE | airfoil leading edge |
| TE | airfoil trailing edge |

**Abbreviations**

| | |
|---|---|
| BEM | Blade Element Momentum |
| CFD | Computational Fluid Dynamic |
| CST | Class/Shape Function Transformation |
| IDDES | Improved Delayed Detached Eddy Simulation |
| HAWT | Horizontal Axis Wind Turbine |
| NSGA-II | Non-dominated Sorting Genetic Algorithm-II |
| NREL | National Renewable Energy Laboratory |
| PSO | Particle Swarm Optimization |
| RANS | Reynolds Average Navier–Stokes |
| SST | Shear Stress Transport |
| TSR | Tip Speed Ratio |

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
