# Peer review of "Multi-Megawatt Horizontal Axis Wind Turbine Blade Optimization Based on PSO Method"

_aerospace, doi:10.3390/aerospace10020158_

Round 1

Reviewer 1 Report

The manuscript entitled A New Blade Optimization Method for Multi-Megawatt Horizontal Axis Wind Turbines and Its Verification using Improved Delayed Detached Eddy Simulation has been reviewed. In this research, PSO method is proposed, and its effects on the enhancement of the power generation of NREL WP-Baseline 1.5 MW horizontal axis wind turbine are investigated, and the algorithms, tunings, and techniques adopted in the present study were general, the presented method can be used as a systematic approach for the aerodynamics shape optimization of multi-megawatt HAWTs. From my point of view, the PSO methodology is relatively novel in aerodynamics, but there are some major issues should be improved in this manuscript.

My comments are listed as follows:

1.       Please read over your manuscript. " Error! Reference source not found" is displayed throughout the half of the article.

2.       In Sec 3.1, what is 'the best shape function is of 7th order' calculated form? How is The Error value in Table 1 obtained? Is The Best Shape Function Order obtained by Mean Error of Error analysis? In Figure 1, there are several curves near 7 'th that are very close to each other. What method is used to obtain this conclusion

3.       Sec 4.2, below Figure 2,” In this study, HAWT power output is the objective function (equal to food availability) and the number of birds (variation of the blade geometry) affects the power enhancement. Should be changed to” In this study, HAWT power output is the objective function (equal to food availability) and the variation of the blade geometry (number of birds) affects the power enhancement. Such bird and food metaphors should be in parentheses.

4.       Is there a specific relationship between the speed and wind speed in the article below Table 6 and Figure 23? If not, how do we figure out Rotational speed?

5.       Figure 25.26.27.28 is more easily distinguishable when compared side by side

6.       The number of the picture described in the article seems to be completely staggered with your picture label. Please note that the label corresponds to each other.

7.       Table 3 can list just the Blades that will be mentioned in the article.

8.       The comparison between optimized power and original power can be added in Figure 30.

9.       The numerical method introduced in the paper is insufficient. Is the code an in-house or commercial (CFX or FLUENT?). What about the time step of IDDES? What about the integral time for the averaged parameters?

10.   For a paper of optimization, the main contribution is to give an optimized direction for the readers, so how to improve the performance of the blades? The authors should offer some major disciplines. A deep explanation of these disciplines based on the optimized results will be better.

11.   The title “A New Blade Optimization Method for Multi-Megawatt Horizontal Axis Wind Turbines and Its Verification using Improved Delayed Detached Eddy Simulation ” is too long and the major points of the paper is not focused. “Multi-Megawatt Horizontal Axis Wind Turbine Blade Optimization Based on PSO method

Author Response

Dear reviewer

Thank you for your time and your valuable comments.

Sincerely yours

Authors

Reviewer 2 Report

This paper implemented CST parameterization and PSO for wind turbine design, followed by CFD validation. Results showed that the optimized blade produced more power at all wind speeds above 4.5m/s.

This reviewer has the following comments.

1. In the abstract, the authors claimed "a Particle Swarm Optimization (PSO) method is proposed", what is the novelty in the PSO the authors proposed?

2. When talking about gradient-based optimization, the authors claimed "In addition, they are very sensitive to initial conditions", which is too strong and should be rephrased.

3. The authors preferred CST over the other parameterization. Are there any downsides of CST?

4. When validating CST method, the authors compared with only three different airfoils, is this sufficient?

5. Figure 1 and Fig. 2 are messed up with each other. And a few other figures or tables were not appropriately referred in the paper. Please fix them.

6. In Table 1, what are the mean and max errors? The authors should show the accuracy on the airfoil coordinates too.

7. During the optimization, the authors should give a clear optimization formulation, with the objective function, constraints, ect. in a formula or table. For the current cases, did the authors only use one design variable?

This reviewer appreciates the authors' efforts. Looking forward to your reply.

Author Response

Dear reviewer

First of all, please accept our best thanks and appreciation for you.

Regarding your valuable comments, we would like to inform you that have tried our best to address all of the comments in detail and point to point. Indeed, thanks to the highly professional and priceless comments, we believe that the level of the manuscript is now elevated considerably.

Best regards,

Authors

Round 2

Reviewer 1 Report

The authors have made sufficient modification according to the comments, the writting quality is improved. I'd like to recommend to accept the paper in the current form.

Reviewer 2 Report

Thanks for the authors' reply and efforts. The authors have fixed all points in my previous review. I suggest this paper be published in this journal.